# Nurturing Faith and Enlightening Minds: Assumptionist Education in the Ottoman Empire

Ediz Hazir [1,2]

1 Department of Russian and East European Studies, Institute of International Studies, Faculty of Social Sciences, Charles University in Prague, U Kříže 8, Jinonice, 158 00 Praha, Czech Republic; ediz.hazir@fsv.cuni.cz or e.hazir@rug.nl
2 Department of Christianity and the History of Ideas, Faculty of Religion, Culture and Society, University of Groningen, Oude Boteringestraat 38, 9712 GK Groningen, The Netherlands

**Abstract:** The text explores the educational activities of French Roman Catholic missions in the nineteenth century, as they evolved from serving local Catholic needs to becoming crucial assets in advancing France's religious–cultural influences and the Holy See's efforts to unify Eastern Christian Churches under Rome. Focused on the Mission d'Orient, initiated during Pius IX's papacy, this study delves into the Assumptionists' educational activities in the Ottoman Empire (1863–1914), which aimed to inculturate the Christian communities of the Ottoman Empire, achieve union with Rome, and build a bridge of knowledge between the Ottoman Orient and Europe. Employing a transnational historical approach, this research utilizes primary sources from the Holy See and the Assumptionist Order, examining religious and educational interactions with Ottoman millets. This article argues that Assumptionist institutions succeeded in inculturation and acted as bridges for cultural exchange. The context includes the French protectorate of the Ottoman Empire, the protégé system, and the Capitulations of 1740, demonstrating the Holy See's use of political and religious alliances. The Assumptionists, influential in advancing the Holy See's interests, are studied regarding their engagement in France and the Orient. Despite valuable insights from existing research, this article seeks to fill gaps by using Assumptionists as a case study, exploring the specific impacts of their education on various religious groups within the context of France's religious–cultural imperialism.

**Keywords:** education; Roman Catholic missions; Ottoman Empire; France; the Vatican





## 1. Introduction

Starting from the mid-19th century, Catholic and Protestant missionaries operating within the confines of the Ottoman Empire demonstrated a dual commitment to advancing both educational and healthcare initiatives. This commitment was manifested through the deliberate allocation of a substantial portion of their personnel and resources towards the establishment of a diverse array of institutions. These included educational establishments, such as schools, seminaries, and universities, as well as medical facilities, like dispensaries and hospitals.

During the same historical epoch, French Roman Catholic missions underwent a noteworthy transformation. Initially oriented towards addressing the needs of local Catholics, these missions underwent significant evolution, emerging as pivotal components in the Holy See's endeavors to consolidate the Eastern Christian Churches under the authority of Rome. The missionary endeavors in education contributed to the forging of channels of communication and cooperation between Eastern Christian populations and the Western sphere. Furthermore, they functioned as conduits for the dissemination and assimilation of European culture, particularly through the influence wielded by the French.

This study examines the educational activities of the Augustinians of the Assumption (the Assumptionists), a French Roman Catholic order, aimed at the inculturation of Ottoman

Christian communities, to establish an environment conducive to achieving the union of Eastern Christians with Rome. Specifically focusing on the Assumptionist presence in the Ottoman Empire, from the start of the Assumptionists' Mission d'Orient in 1863 to the expulsion of the French missionaries from the Ottoman Empire in 1914, this article seeks to address the following question: What was the Assumptionists' strategy in achieving the inculturation of Ottoman communities to the service of the Holy See's unification project? Through this exploration, I argue that the Assumptionist educational institutions achieved success on two fronts. Firstly, they accomplished the inculturation of various Ottoman communities, operating within an environment where the *Mission d'Orient* was perceived by indigenous communities as Rome's proselytism. Secondly, these institutions acted as bridges for cultural exchange between Europe and the Ottoman Orient.

This article employs a transnational approach to comprehensively analyze the relationships between Assumptionist activities and France, the Holy See, and the Ottoman Empire. Many historians of international relations argue that transnational history primarily focuses on the actions of non-state actors (Saunier 2013, p. 32). In this paper, the specific focus is on the Assumptionists' religious and educational interactions with various Ottoman millets. Therefore, a transnational historical approach is deemed the most appropriate method through which to analyze these interrelations.

To conduct this research, the author utilizes content analysis for both primary and secondary sources. The primary sources were obtained from the online archives of the Vatican, as well as the Assumptionists' Library in Kadıköy. Additionally, this paper benefited from the *Annales de la Préparation de la Foi* and the Assumptionists' magazines, including *Echos d'Orient*, *Mission des Augustins de l'Assomption*, *L'Œuvre de Vocation* in *Echos du Noviciat de Notre Dame de l'Assomption*, and *La Croix*.

The Assumptionists, a French Roman Catholic congregation, were deeply influenced in their activities in the Orient by the evolving developments in France, which encompassed religious, cultural, political, and economic relations with both the Ottoman Empire and the Holy See. The Capitulations of 1740, which laid the foundation for French legislation within the Ottoman Empire, were granted by the Sublime Porte, allowing France to exert its influence under the pretext of safeguarding the Holy Places, as well as the Catholics residing within the Ottoman Empire (Ghaleb 1913). Similar to its predecessors, the Capitulation of 1740 acknowledged the authority of French consuls in protecting individuals without French nationality, extending this protection to Ottoman subjects under specific conditions. Under the French protectorate, French citizens not designated as Ottoman subjects retained their nationality, while being subject to French jurisdiction, reflecting their legal status in France. Ottoman subjects, on the other hand, enjoyed advantages similar to those granted to foreigners under this protectorate. A diverse group of non-Muslim subjects of the Porte willingly opted to escape Ottoman jurisdiction and fall, instead, under consular authority (Cirilli 1898, pp. 237–39).

Following the Treaty of Paris in 1858, the Ottoman Empire controlled territories marked by complex politics, arising from diverse ethnicities, languages, and religions, while the Great Powers competed over the "Eastern Question". The religious–cultural aspect of this competition aimed to gain the support of local populations. Christian missions acted as representatives of the Western powers in this endeavor, as they had direct interaction with the communities.

The culmination of the Papal States in 1870, following significant territorial reductions in 1860, marked a considerable decline in the political sway of the Church across Europe. The Vatican strategically employed the Capitulations, the protégé system, and the French protectorate of Catholics to advance its interests, increase its influence over Eastern Christians, and seek the unification of the Orthodox and Roman Catholic Churches. The rivalry between the Great Powers over the Holy Places, combined with Russian influence over the Orthodox community, British–American attempts to spread Protestantism among Ottoman Christians, and French efforts to maintain their protectorate, resulted in a clash of denominations in the Ottoman lands. Despite conflicts with the Third Republic, the Holy

See leveraged French political and religious support to increase its influence over Ottoman Christians until the first quarter of the twentieth century. Ultimately, the success of Roman Catholic missions in the Orient depended on the collaboration between France and the Holy See, as well as their interactions with the Sublime Porte (Hajjar 1979).

The Assumptionists emerged as one of the leading Roman Catholic orders, advancing the Holy See's interests during a period when European powers were primarily focused on protecting holy places and utilizing education to gain cultural and religious advantages over each other. Their mission in the Orient, spanning the territories of the former Russian and Ottoman Empires, exemplified their profound engagement with indigenous communities, impactful educational endeavors, and dynamic interactions with various states.

Previous research on the Assumptionists' and Oblates' Mission d'Orient provides valuable insights into the strategies employed by both the French and the Vatican to further their interests. The research, primarily conducted by the Assumptionists themselves, is complemented by contributions from multiple scholars to the historiography of the Mission d'Orient. Firstly, the Assumptionist mission can be extracted from broader research that offers an understanding of the political environment where the mission was born (e.g., Hajjar 1979, pp. 134–549; Prudhomme 1994, pp. 295–325; and Fouilloux 2000, pp. 167–75). Secondly, scholars who focused on the Assumptionists provided a thorough analysis of Roman Catholic missions, as well as the reasons behind their transformation in the nineteenth century (e.g., Babot 2000, pp. 13–117; Vrignon 2007, pp. 83–138; and Babot 2011, pp. 9–46). Thirdly, the research primarily focuses on the overall activities of the Assumptionists and explores Assumptionist Education as part of the overview (e.g., Jacob 2000, pp. 241–321; Babot 2000, pp. 327–83; and Thobie 2009, pp. 593–674).

While benefiting from the previous research on the subject, my objective is to utilize the Assumptionists as a case study through which to comprehend the role of French Roman Catholic missionary education as a vehicle for the spread of Western culture and ideas through the French language, while creating bridges of communication and cooperation between the Christian churches of the East and the West. Moreover, I intend to investigate the impact of Assumptionist education on various religious and cultural groups, including Catholics, Armenians, Greeks, and Muslims.

## 2. Impact of Political Diplomacy, Missionary Endeavors, and Indigenous Engagement

The Ottoman Empire employed a complex system, called the millet system, which consisted of various religious and administrative structures. The purpose of the millet system was to ensure religious and cultural autonomy for non-Muslim communities within the Ottoman Empire, thereby promoting stability and harmonious coexistence. It effectively enabled the empire to manage its diverse population by granting a level of self-governance to various religious and ethnic groups. The Greeks obtained the status of millet in 1453, while the Armenians were recognized in 1461. These two Christian communities were main targets of the Assumptionist mission, as the conversion of Muslims was not an option, and the conversion of Jews was unlikely.

The Tanzimat reforms (1839–1876) were a turning point for the Ottoman millets, as they promised equality to all Ottoman subjects, while the Ottoman Reform Edict of 1856 assured that "all the privileges and spiritual immunities of the churches would be respected and that individual Christians would enjoy all civil rights on the same level as Muslims" (Frazee 1983, p. 225). The Ottoman Westernization reforms resulted in a flow of European Christians arriving to benefit from the freedom that the Tanzimat offered.

The pro-Western transformation of the Ottoman Empire in the nineteenth century had a positive outcome for the Christian missions. Christian missionaries in the Middle East prioritized education and health initiatives, allocating a significant portion of their personnel and resources to establish schools, colleges, universities, dispensaries, and hospitals. Despite the charitable focus of these institutions, aimed at addressing the needs of the economically disadvantaged, their services extended beyond the most marginalized segments of the local population. In addition to educational and health-related efforts, mis-

sions actively participated in explicit charitable activities, including operating orphanages, providing shelter for prostitutes, visiting prisoners, and supporting unskilled workers. The missionaries' commitment to both educational and charitable endeavors reflects their comprehensive approach to improving the well-being of the communities they served (Verdeil 2020, p. 23).

The Roman Catholic missions had been active within the Ottoman Empire since the seventeenth century, primarily focused on serving the needs of European Catholics. However, following the Crimean War (1853–56), a collaborative effort between the Holy See and France sought to intensify their religious and cultural engagement in the Ottoman Orient. The French Roman Catholic missions emerged as pivotal players in advancing the expansion of French religious and cultural influence, while serving as representatives for both France and the Vatican.

The Augustinians of the Assumption, a French Roman Catholic Congregation, was founded in Nîmes by Emmanuel d'Alzon in 1845. The name of the congregation came from St. Augustine of Hippo, who was considered a "friend of the Orient", due to his knowledge of the Greek language and esteem of the great Greek philosophers (Salaville 1931) (Salaville 1922, pp. 382–93). The "*Mission d'Orient*" began for the Assumptionists on 4 June 1862, with Pope Pius IX's famous sentence, "I bless your deeds in the Orient and Occident", addressed to Fr. D'Alzon. Pius IX knew of Fr. D'Alzon's aspiration to serve the Christians in the Middle East, which had grown stronger after 1860, when France militarily intervened to safeguard persecuted Maronite Christians from the Druzes. Therefore, he assigned the Assumptionists to this mission, which was initiated in the Balkans. The Mission d'Orient had both a religious goal and a political goal. The religious goal was "The elimination of the "Photian Schism (Vailhe 1934, p. 349)"". The political goal was to assist France in diminishing the Russian influence over the Ottoman Orthodox (Hazir 2023a).

The Assumptionist mission in the Ottoman lands started with Fr. Victorin Galabert's arrival in Istanbul on 10 December 1862. However, following the Holy See's instructions, he relocated to Plovdiv in 1863 (Monsch 2000, p. 125), and then to Edirne to establish the first Assumptionist school in 1867 (Fleury 2000, p. 115). The Oblates of the Assumption arrived in Edirne on 7 May 1868, to support the Assumptionists (Jacob 2000, p. 243). Initially, the target community for the mission was the Balkan Slavs, as Bulgarian Christians already had the intention to form closer relations with the Holy See to detach themselves from the patriarch in Phanar and Russian pan-Slavism (Voillery 1980, pp. 31–47). The detachment of Bulgarians from the Orthodox Patriarchate in 1870, and the success of the unionist movement, encouraged the Assumptionists to expand their activities. After the Russo-Ottoman War of 1877–78, which resulted in Ottoman defeat and Bulgarian independence, the Assumptionist focus turned to Istanbul and Anatolia, even though they continued their activities in the Balkans.

The Assumptionists, along with their female counterparts, the Oblates (founded in 1865), advocated for the inculturation of Eastern Christians and the inclusion of the Muslims and Jews to ensure the success of their mission. The term "Eastern Christians" encompassed a wide range of different churches. Eastern Orthodox Churches, Maronites, Melkites, Armenians, Western Syrians, Eastern Syrians, and Copts were represented in the Ottoman Empire (Amsler 2020, p. 190). The Assumptionnists and Oblates emphasized the importance of quality education and free healthcare in this endeavor. Furthermore, through their educational institutions, they aimed to build an understanding of the Eastern Christian populations, especially the Greeks and the Slavs, while exploring their cultures, traditions, languages, and history. The mission served as a vital conduit for cultural exchange between Europe and the Ottoman Orient, facilitating the exchange of ideas, knowledge, and experiences. This cultural bridge significantly contributed to fostering mutual understanding and establishing a platform for dialogue between the East and the West.

For instance, in his report, Monsignor Zschokké, an Austrian prelate who served as the director of the Austrian hospice in Jerusalem in 1897, praises the French Catholic

activities in the Holy Land. "I was not at all surprised to see everything that France has been able to accomplish in Jerusalem. The board of the Austrian hospice struggles to maintain 70 beds, while the French Catholics continuously establish one charitable work after another in the holy city of Jerusalem. Not to mention the sacrifices they make at home in support of Catholic schools and universities in their own country. It is these very same French Catholics who send missionaries and nuns around the world, using French funds to build churches, religious houses, and schools everywhere. Catholic France has remained, until our days, the eldest and most devoted daughter of the Church. A nation that makes such sacrifices for the Catholic Church can surely rely on divine Providence, which will undoubtedly grant it a better fate one day (Bailly 1897)".

Thus, in terms of supporting the Catholic missions, the Republic did not act in the name of a divine mission, but in that of a national civilizing mission (Prudhomme 2008). The Assumptionists' connection to the French state was as strong as their connection to the Holy See. Thus, their mission did not only export Catholicism, but also the ideas of enlightenment, secularism, human rights, and republican rights, through the power of French as a language. "France's educational mission was not exercised triumphantly or monolithically. It was the fruit of learning lessons from previous experiences, competition with other denominations, and dealing with conflicting ideologies. For example, Catholics relied on secular education to maintain their influence when the conversion of the locals was not an option (Dana and Cabanel 2007)".

The Assumptionist activities in Ottoman Anatolia noticeably increased after the Treaty of Berlin in1878, as the Ottoman Empire lost the majority of its territories in the Balkans. After the death of Fr. d'Alzon (1880), François Picard became the head of the order. Under Fr. Picard's supervision (1880–1903), Assumptionist establishments multiplied in Istanbul and Anatolia. His good relations with Leo XIII, and the change in the Holy See's Eastern politics, started a new chapter for the Assumptionists. In the Orientalium Dignitas on the Eastern Churches published in 1894, Leo XIII described the importance of the Eastern rites for Rome (Leo 1894). He repeated his hope for the union, while instructing Catholic missionaries on their approach toward the Christians of the Orient. Leo XIII slightly changed the Vatican's oriental politics, in terms of ending Latin proselytism to reach acculturation and the idea of the global union as mentioned in the Councils of Lyon and Florence (Fouilloux 2000, p. 73). The Jerusalem eucharistic congress in 1893 described the Holy See's position regarding a unified Church. The greatest obstacle preventing the union was the Vatican's proselytism. The Roman Catholic clergy ensured that the Eastern Catholics, united under Rome, freely practiced their rites and traditions. During the congress, All Catholic Churches celebrated the Holy Sacrifice according to their rites. Chaldeans, Syrians, Maronites, Copts, Greek-Melchites, Greco-Slavs, and Armenians, in turn, offered the Eucharist, and the Orthodox Greeks, who were welcome to participate, could see with what respect and with what sympathy the Latins attended these services in various languages and following various rites (Bailly 1897, pp. 33–38). "Pius X unlike Leo XIII did not go back on the policy of support for the Uniats of the East. He also wanted to revive their life, encourage their Eastern liturgies, foster their national traditions, and abandon the Latinizing endeavor (Chadwick 1998, p. 546)".

Diplomatic reconciliation among the Holy See, France, and the Ottoman Empire created a favorable environment for Roman Catholic missions operating in Ottoman territories until the First World War. This understanding fostered tolerance and cooperation, allowing missions to access Christian communities within the empire. Improved relations contributed to a more conducive environment for the propagation of the Catholic faith and missionary endeavors. However, the ultimate success of these missions relied on the efficacy of missionary activities and the ability to cultivate positive relationships with indigenous populations, with the Greek and Armenian Patriarchs having decisive positions under the Sublime Porte, due to their authority within their respective millets.

The impact of Assumptionist education on Catholics, Armenians, Greeks, and Muslims was generally positive, which is attributed to the quality of education and the equal

opportunities provided to these communities. The conflicts that arose between the Assumptionists and the communities they served were mainly political. Greeks, Armenians, and Muslims responded to Assumptionists' activities in light of this tension.

The primary conflict with the Orthodox communities was the assertion of Papal infallibility. Papal infallibility presented a challenge to achieving unity with the Orthodox, as Eastern Christians rejected Rome's claim of superiority. This matter was deliberated during the First Vatican Council (1869–1870). Moreover, since the recognition of the Armenian Catholic millet by the Sublime Porte, the Ottoman government saw Western attempts to influence its non-Muslim subjects as interference in its internal affairs.

The Assumptionists' engagement with non-Catholic communities varied, depending on factors such as the millet (Greek, Armenian, or Muslim), their position within the hierarchy (Church or believers), and the geographical context. Generally, Assumptionists interacted with Muslims through state authorities, with construction, renovation, and foundation projects being permitted by the local governor upon obtaining the necessary permissions.

Interaction with Armenians proved to be relatively easy, compared to that with the Greek Orthodox, largely due to the presence of an Armenian Catholic millet. However, engaging with the Greek Orthodox posed challenges. For example, when the Assumptionists sought to build a bell tower for their Church in Eskisehir, local Greeks attempted to obstruct the construction by filing complaints with the local authorities. Father Bertin described the Greek attitude as follows: "The jealousy of the Orthodox and their deadly hatred had caused us many troubles until now, and this time too, it was expected that the difficulties would arise not from Muslims, but from the Greeks."[1] In the end, the bell tower was built with the aid of the Armenian and Greek students of the congregation.

Missionaries' efforts to communicate Catholic Church teachings and provide education, healthcare, and humanitarian assistance were the leading factors in gaining acceptance and support from local communities. The missionaries' ability to understand and respect the cultural, social, and religious contexts of Ottoman Christians was essential in fostering meaningful connections and building trust (See Hazir 2023b, pp. 8–14).

Between 1867 and 1914, Assumptionists established various institutions, from Edirne to Trabzon. However, the onset of the Balkan Wars (1912–13) led to their decline, influenced by significant demographic changes in Anatolia, which was driven by Turkish nationalism promoted by the Committee of Union and Progress, the influx of Muslim refugees from the Balkans, and the 1913 population exchange involving the Ottoman Empire, Greece, Serbia, and Bulgaria. The Committee of Union and Progress unilaterally abolished capitulations in 1914, signaling the end of the French protectorate of Roman Catholics. All French institutions, except for churches, hospitals, and orphanages, were closed, and French missionaries who did not enlist with France for the war were expelled.

### 3. Enlightening Minds

*3.1. Characteristics of Assumptionist and Oblate Education*

The Assumptionists and the Oblates directed their efforts toward serving diverse Ottoman communities. Although their initial endeavors were concentrated in France, they referred to the "*Mission d'Orient*" as their "other lung". In the Orient, their commitment centered on promoting Roman Catholicism through a variety of activities, with a particular emphasis on education.

According to Fr. D'Alzon, the only path to reach the union was through education. In *Le Directoire* (1859), he wrote, "Teaching is one of the most powerful ways of fulfilling the desire to make the Kingdom of Jesus Christ come." (Guissard 2002, p. 64) Fr. d'Alzon believed that, first, it was necessary to build and educate a local clergy. Second, the congregation had to find educational institutions with Western curricula, while respecting the indigenous traditions and values. Third, the congregation had to attend to the locals' needs (the foundation of dispensaries and orphanages) without distinction of nationality, ethnicity, or religion.

Missionary education encompassed the instruction of the French clergy about local populations, the Catholic upbringing of the local clergy (specifically, Greek and Slav Catholics), and the education of children. Additionally, the Assumptionists, driven by a passion for knowledge and ultramontane zeal, endorsed modern education.

Educational endeavors by the Assumptionists and Oblates involved establishing seminaries, schools, alumnats (boarding schools), orphanages, parishes for other rites (e.g., Byzantine rites), formation houses (noviciates), and study houses for young religious people. Furthermore, they played a significant role in promoting the dissemination of knowledge through their research and publications on Byzantine and Ottoman culture and history.

The Assumptionists and Oblates served all Ottoman communities, including Latin and Uniate Catholics, Greek Orthodox, Armenian Gregorians, Jews, and Muslims. In addition to their duties in the Balkans and Anatolia, Leo XIII also instructed the Assumptionists to attend to the spiritual needs of Istanbul's Latin Catholics and Greeks.[2] Thus, their schools were multicultural and multinational. They were a mosaic of people from almost all Ottoman communities (i.e., French, Italians, Austrians, Spanish, English, Greeks, and Ottomans (see Thobie 2009, p. 603)).

The Assumptionists knew that the inculturation of different Ottoman millets and the inclusion of girls into education were crucial for the mission's success. Thus, a year after Fr. Galabert's arrival in Edirne in 1868, the Assumptionists founded their first school in the Ottoman Empire. A school for boys was followed by a school for girls, founded by the Oblates in Edirne. In 1869, the mission in Edirne consisted of three schools (St. Vincent (also a dispensary), Our Lady the Helper, and Our Lady of the Pardon), and, in 1871, expanded to include a hospital, a Bulgarian school of Kaik, and two medical facilities. An orphanage was added in 1874, but closed in 1882 because the Ottoman authorities forbade Muslim children from attending (Jacob 2000, p. 243). Both Oblates and Assumptionists offered humanitarian aid to local communities (especially Muslims) during the Russo-Turkish War of 1877–1878 (Zediu 2010, p. 38). In 1880, there were two communities in the Balkans: Plovdiv and Edirne (and its suburbs *Karaağaç* and *Kaïk*) (Perriet-Muzet 2008, p. 12). The Assumptionist educational institutions grew from Edirne to Trabzon until World War I. In 1914, the Assumptionists and Oblates had twenty active establishments with one hundred and fifty clergy members (Thobie 2009, p. 594).

The schooling of the children was a priority for the mission. That is why the foundation of a church was immediately followed by that of a school. The missionaries aimed to reach out to the rest of the population through educating the local children. However, missionary education was a bigger project than just raising Catholic children. Assumptionist education had three stages: the education of the Assumptionists on local populations (traditions, history, and language), the Catholic education of the local clergy, and the schooling of children. Therefore, the clergy acquired sufficient knowledge of the local languages to be able to teach catechism and church history (Perriet-Muzet 2008, p. 6). The first two took place in the seminaries, and the third one in primary and secondary schools, alumnats and noviciates.

*3.2. Seminaries*

The training of local priests practicing Byzantine liturgy and acknowledging the primacy and magisterium of the pope was pivotal for integrating into communities. To achieve this, the first Assumptionist seminary was established in Edirne in 1863. The seminary was crucial in facilitating the French clergy's understanding of local languages, cultures, and traditions. One notable Assumptionist seminary, Saint Leo of Kadiköy, founded in 1895, was particularly active. Situated on Istanbul's Anatolian side, it drew its name from both Leo the Great, the pope of the Council of Chalcedon, and Leo XIII, a proponent of the initiative. The Seminary, established following Leo XIII's instructions, operated prominently until 1914, with brief periods in the 1920s and 1930s. Internal and external conflicts led to the relocation and closure of several seminaries within the Ottoman

Empire. For example, Edirne's St. Peter and St. Paul seminaries moved to Kumkapı, Istanbul, in 1884, and later to Fenerbahçe, in 1896 (Jacob 2000, p. 245).

Seminaries served as the cornerstones of missionary education, acting as bridges between the Ottoman Orient and the West. Their extensive research and publications on Byzantine, Slavic, and Anatolian cultures facilitated their integration into Ottoman society. On 7 October 1895, the Assumptionists established the Center for Oriental Studies in Istanbul, which later evolved into the Institute of Byzantine Studies (Faillier 1995, p. 5). Leo XIII specifically tasked the Assumptionists with exploring the history, culture, and traditions of the Ottoman population (Picard 1897). The primary reason for establishing the Kadıköy seminary was to fulfill Leo XIII's desire to facilitate the return of dissidents from the Orthodox Churches to the Roman Catholic Church and serve the local Latins. In May 1902, the clergy in the seminary served fifty-five Latin and eighteen Greek Catholic families.[3] The Assumptionists focused on training Greek clergy in Kadıköy, where indigenous priests practiced Byzantine liturgy in Greek, while teaching the public about the Pope's primacy and authority. Achieving this goal required instructors with deeper connections to the local populations and profound understanding of the history, language, and liturgy of the Eastern Churches, particularly the Greco-Slavic Churches. Many French Assumptionists obtained the Byzantine rite and spoke Greek in their daily lives for this purpose (see Figure 1).[4]

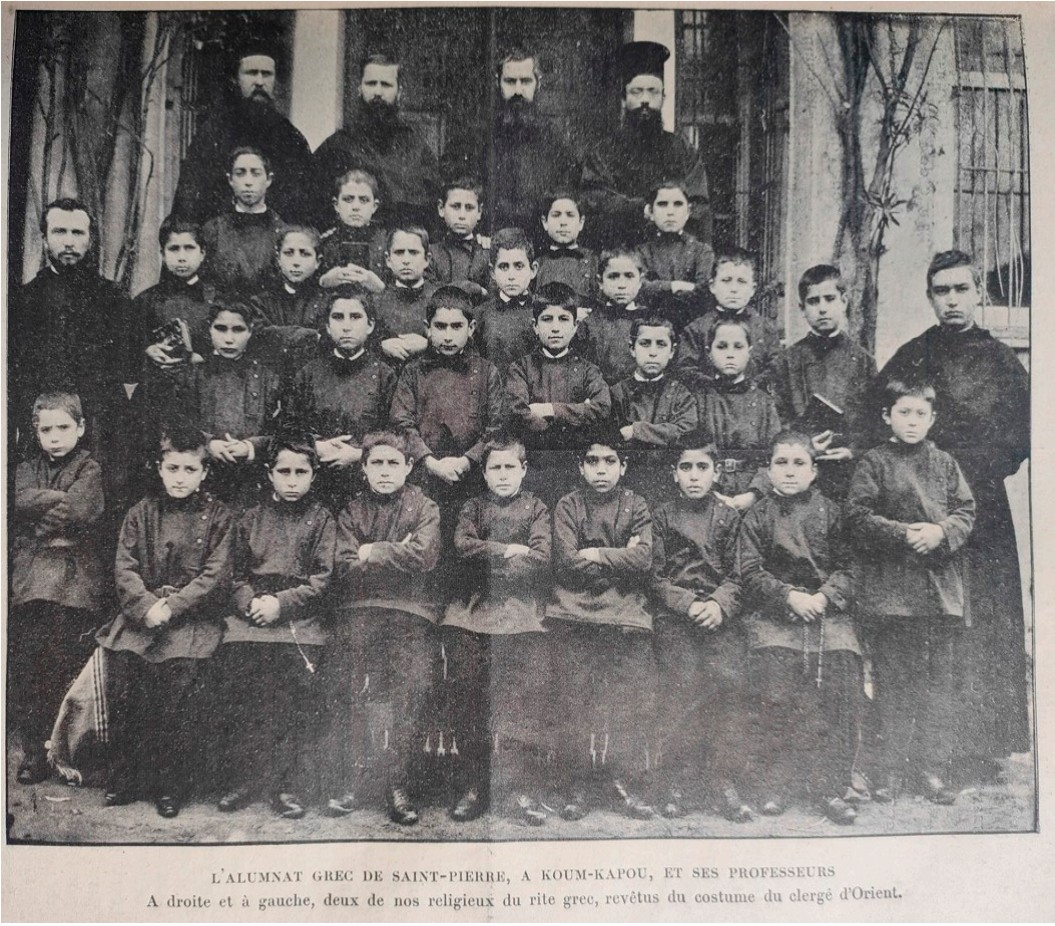

L'ALUMNAT GREC DE SAINT-PIERRE, A KOUM-KAPOU, ET SES PROFESSEURS
A droite et à gauche, deux de nos religieux du rite grec, revêtus du costume du clergé d'Orient.

**Figure 1.** The St. Peter Greek Alumnat in Kumkapi and its instructors. The instructors of the Greek rite are dressed the same as the Orthodox clergy[5].

Initially functioning as a college of professors responsible for educating priests from these communities, the newly formed institution did not include the publication of clergy research in its initial plan. However, this quickly became central to the seminary's endeav-

ors (Faillier 1995). Furthermore, instructors within the St. Leo Seminary founded a School for Advanced Studies, dedicated to research and publication.

*3.3. Publications: Échos d'Orient*

In 1897, Father Louis Petit initiated the publication of a magazine focusing on Byzantine/Orthodox history called *Les Échos d'Orient*. The purpose of the work in Kadiköy and the publication *Échos d'Orient* was to foster a deeper understanding of the Christian East, which was deemed crucial because "to be ignorant of the East was to almost be ignorant of the Church." The Assumptionists, editors of the *Echoes de Notre Dame de France*, received a special mission from the Holy See in the East, to work towards the unity of the Churches through the respect and mystical preservation of Christ. This special apostolate called for special studies, full of interest, either in the rituals themselves, or in the peoples and countries that preserved their traditions (Faillier 1995, p. 17).

Initially, the bulletin had a generic title. Still, it later received a more specific and evocative title, *Échos de Notre-Dame de France à Jérusalem* (Echoes of Our Lady of France in Jerusalem), which directly referenced the Assumptionists' model. The Assumptionists held a profound connection to France, deeply rooted in their identity, while also maintaining a delicate balance with their unwavering obedience to the Vatican. The Congress of Jerusalem served as an important platform for discussions on Eastern rites and exemplified the Roman Catholic Church's openness to engaging with the Orthodox Churches. It can be said that the establishment of the seminary in Kadiköy and the subsequent creation and content of the *Échos d'Orient* (1897–1942) were, in a sense, direct outcomes of this influential event.

The purpose of Assumptionist activities and *Échos d'Orient* was articulated by Edmond Bouvy, an Assumptionist and a specialist in Oriental Christianity, who wrote, "The Échos d'Orient is beyond being solely a speculative journal focused on archaeology, history, liturgy, and Byzantine literature. We have a specific and, let it be emphatically stated, a supernatural and apostolic objective: to involve Western Christians in Eastern Christian communities; to actively contribute to the resolution of the schism, as part of the significant unity efforts initiated by Leo XIII; and to catalyze a threefold movement within the Church, particularly in France, encompassing prayer, study, and action in support of the East and Eastern affairs (Bouvy 1898, p. 257)".

In due course, local research on sacred history and archaeology provided them with a newfound impetus that not only adhered to the original program, but also surpassed it. The Assumptionnists, who assumed the role of regular editors for the *Echos d'Orient*, were bestowed with a distinctive mission from the Holy See in the East, to foster the unity of Churches by honoring and safeguarding ancient rituals. This unique apostolic undertaking necessitates comprehensive studies that delve into the captivating intricacies of these rituals, as well as the cultures and nations that diligently preserve their age-old traditions (Bailly 1897, pp. 1–2).

Alumnats and Noviciates (*Maisons d'Etudes*)

The alumnats were boarding schools, through which the congregation prepared their recruitment fostered vocations. Each alumnat welcomed thirty or forty children, who, due to financial constraints, were unable to gain admission to the "Petit" Seminaries. These students spent five years in the alumnats, where they received both religious and literary education, preparing them to apply to the Grand Seminary or a religious institution of their preference. One of the most prominent alumnats was the Alumnat of Phanaraki, established in 1889.[6]

Prospective missionaries received training in the alumnats. For instance, in 1899, the Karaağaç orphanage underwent replacement with a new institution, dedicated to facilitating productivity for missionary activities in the East. This facility operated as an alumnat for young girls from the Orient, enabling them to actively pursue their vocations. Over time, these individuals transformed into dedicated religious emissaries, Christian educators, or nurturing figures within families (Quenard 1925, p. 25).

The novitiate served as the entrance to the congregation, involving a year of religious training that concluded with the initial annual vows of poverty, chastity, and obedience. These vows could be renewed three times, eventually leading to a profound commitment to a perpetual profession. The novitiate welcomed young individuals who had finished their studies in seminaries, ecclesiastical colleges, high schools, and other educational institutions. Upon completion of the novitiate, these young religious individuals embarked on a three-year journey as a scholastic dedicated to philosophy, followed by an additional four years as a theological scholastic, as part of their preparation for the priesthood (Quenard 1925, p. 129).

A graduate of the alumnat aspiring to enter the novitiate did so towards the conclusion of their literary studies and remained there for two years. Following the novitiate period, these individuals pursued studies in philosophy and theology, either in Rome, Toulouse, Jerusalem, or Kadıköy.[7]

*3.4. Schools*

The Assumptionists held a strong belief in the significant role of schools, catering to both boys and girls, within Roman Catholic missions. Their primary objective was "to present the Catholic truth in its entirety, devoid of any dilution or exaggeration (Picard 1897, p. 5)". By educating children, they aimed to create pro-Western citizens who could influence the rest of the population, making the school a central element of each congregation. In their pursuit of inculturation, the Assumptionists embraced all Ottoman communities, including Jews and Muslims, recognizing the importance of their presence. However, their close association with France proved to be a drawback in certain cases, as they faced accusations of promoting French nationalism. Consequently, they redirected their focus toward teaching and distanced themselves from their national origins. They understood that any form of discrimination or comparison between Eastern and Western cultures would jeopardize their mission. Moreover, they actively sought support from local (Greek, Slavic, and Armenian) teachers, who could provide guidance in effectively engaging with the local population.

In 1897, the Assumptionist schools offered four years of study, intending to add a preparatory and a fifth class. The school curriculum consisted of five main areas: Religious Education, Classical Education, Schedule, Discipline, and Pedagogical Matters. Religious Education consisted of Catechism, the History of the Church—including the history of each community—practices of piety, and spiritual exercises (Muslims and Jews were exempt from religious practices). Classical Education's motto was "Don't touch various materials just to mention them but teach what should not be ignored!" Thus, it included Reading and Writing, with French as the primary language, Other European languages and the local ones (Greek, Armenian, and Turkish), Geography, Sciences, Painting, Singing, Gymnastics, and Courtesy (See Table 1).

The Assumptionist schools were as multicultural and multinational as their parishes were. For instance, Architect Georges Radet describes Eskisehir's school as "sixty children of all religions and all nationalities: Catholics, Protestants, Orthodox Greeks, Gregorian Armenians, Muslims, Jews and a European group of French, Germans, Italians, Austrians, Montenegrins (Jacob 2000, p. 268)." Another example was the school in Kumkapı, Istanbul, which had one hundred and sixty students in 1902[8], distributed as follows: 37 Catholics of different rites (23.12%), 64 Greek Orthodox (40%), 29 Gregorian Armenians (18%), 2 Protestants (1.25%), and 28 Muslims (17.5%) (Jacob 2000, p. 276).

The Oblates played a significant role in the mission's success by providing education, establishing strong connections with local communities through dispensaries, and collaborating with other Roman Catholic orders, such as the Jesuits, during their mission to Little Armenia between 1881 and 1914. For instance, in 1901, the Haydarpaşa parish served over four hundred Catholics, the school had two hundred and twenty students, and the dispensary attended to two thousand sick people, with only thirteen Oblates.[9] They were positioned to take on a more proactive role in the congregation's activities in the Orient,

engaging in language studies and guiding Eastern Catholic students to participate in their respective church rites (Picard 1903, p. 8).

**Table 1.** Number of hours per class and various subjects (Picard 1897, p. 24).

| Matières d' Enseignement | 1re Classe | 2e Classe | 3e Classe | 4e Classe | 5e Classe |
|---|---|---|---|---|---|
| Instruction Religieuse | 2h. ½ | 2h. ½ | 2h. ½ | 2h. | |
| Français et Enseignement Scientifique | 9h. ½ | 9h. ½ | 8h. ½ | 8h. ½ | |
| Calligraphie | 2h. ½ | 2h. ½ | 2h. ½ | 1h. ½ | |
| Deux Autres Langues | 10h. | 10h. | 10h. | 10h. | |
| Histoire | 2h. | 2h. | 2h. | 2h. | |
| Géographie | 1h. ½ | 1h. ½ | 1h. ½ | 1h. ½ | |
| Arithmétique | 4h. | 4h. | 5h. | 5h. | |
| Tenue des Livres | - | - | - | 1h. ½ | |
| Dessin | 1h. ½ | 1h. ½ | 1h. ½ | ½ h. | |
| Chant | 1h. | 1h. | 1h. | 1h. | |
| Gymnastique | Pendant | les | récréations. | | |
| Politesse | ½ h. | ½ h. | ½ h. | ½ h. | |

Moreover, the Oblates significantly contributed to the promotion of a relatively novel concept in the Ottoman lands through the establishment of schools for girls. This action challenged prevailing norms concerning female education and the role of women within Ottoman society. Father Leonce, in his observations in Karaağaç, noted a contrast in marriage customs between France and Turkey. In France, it was customary for a young woman to bring a dowry when proposing marriage, enhancing the financial status of her prospective life partner. However, in Turkey, a man typically purchased his wife, a practice also observed among Christians. Young individuals in Turkey willingly engaged in extensive and challenging labor, subjecting themselves to various sacrifices, all aimed at accumulating the necessary funds to secure a companion.[10]

The establishment of girls' schools during that period was an uncommon initiative within local communities. Father Marcelin Guyot, the first Assumptionist in the future capital of Turkey, Ankara, described the prevailing opinion regarding girls' education in the Orient as considering it useless (Jacob 2000, p. 273). Despite skepticism, the Oblates overseeing these schools achieved notable success.[11] Examples include the Haydarpaşa School, which, by 1896, boasted a diverse student body representing various nationalities and religions, with 125 female students (Thobie 2009, p. 658). Additionally, the Oblate School of Sainte-Hélène in Edirne predominantly enrolled Armenian students, including the daughter of the Pasha (Jacob 2000, p. 275). The local communities' interest in the Assumptionist and Oblate schools stemmed from the perceived quality of education. Some viewed it as a pathway to the West. For the Assumptionists, teaching Catholics and "dissidents" in the same schools was deemed necessary, as the interaction between different cultures facilitated a bond under Catholicism.

Therefore, as Catholicism was not under the rule of any nation, the teachers were instructed to avoid promoting patriotism and comparison between the West and East. That is why it was essential to focus on teaching languages and national histories instead while giving the schools a less exclusive French look, as French schools meant "foreign" schools (Picard 1897, p. 6). The Assumptionists understood that avoiding any nationalist tendency was a must for the Mission's success. In the education program of the Assumptionist schools, redacted by Fr. Picard in 1897, the reasons for the mediocre results obtained until then were cited as follows: the religious forgot that they are equally apostles and educators, the lack of knowledge in native languages, the absence of Churches in the Eastern rites, the

advanced age of the children, and giving the schools a national character (Picard 1897, p. 7). Moreover, discrimination by calling the Eastern Christians schismatics was not acceptable under any circumstances, as "having a real connection with the students teaches them the love of their nation".

## 4. Conclusions

The Augustinians of the Assumption played a significant role in providing high-quality French Roman Catholic missionary education, with various objectives, within the Ottoman Empire. France utilized missionary education as a means to advance its religious and cultural influence over the Ottoman Christians. The Holy See also recognized the value of this education in uniting non-Catholic Ottoman Christians, including Armenians and Greeks, under Rome. One of the key factors contributing to the success of the Assumptionists was their approach to education, which respected indigenous cultures and traditions. By integrating European modernity into a multiethnic and multireligious society, they aimed to enhance communication among diverse groups and emancipation within their institutions. The education received in Assumptionist schools, irrespective of religious affiliation, laid a valuable groundwork for nurturing individuals inclined towards Western values and offered promising prospects within the multicultural and multinational society of the Ottoman Empire. Unintentionally, this educational system played a role in fostering the development of national identity, as republican ideas eagerly contributed to this undertaking.

The findings of this study demonstrate significant achievements by the Assumptionist educational institutions in two key areas. Firstly, they successfully facilitated the integration of diverse Ottoman communities, particularly in an environment where the *Mission d'Orient* was perceived by indigenous communities with national inclinations as Rome's proselytism. The Assumptionists effectively navigated these complexities and managed to create an inclusive atmosphere conducive to the desired union. Moreover, they contributed to the inclusion of girls into communities by educating them.

Secondly, the Assumptionist educational institutions served as vital conduits for cultural exchange between Europe and the Ottoman Orient. Through their presence and educational initiatives, they facilitated the exchange of ideas, knowledge, and experiences, thereby contributing to a deeper understanding and appreciation of different cultures. They were especially well known for their excessive and detailed research on Byzantine culture and history. The bridge of culture they attempted to build played a significant role in fostering mutual understanding and establishing a platform for dialogue between the East and the West. The publication of *Echos d'Orient* was the ultimate proof of this dialogue.

Overall, the Assumptionist and Oblate missions, through their educational activities to achieve the objectives of the Holy See's unification project, demonstrated their ability to navigate complex sociopolitical dynamics and successfully fostered an environment where indigenous communities could embrace their own cultural heritage while forging closer ties with Rome. A Roman Catholic mission was completed when a cemetery was added to the mission's church, hospital, and schools. Despite the Mission d'Orient not reaching its ultimate goal, the ongoing Assumptionist presence in Turkey shows that the Assumptionist and Oblate efforts in facilitating cultural exchange between Europe and the Ottoman Orient enriched the broader dialogue between civilizations, while their research and publications contributed to the region's historiography. Moreover, their efforts resulted in a rapprochement between the Eastern and the Western Christians.

**Funding:** This publication was supported by the SVV project of the Institute of International Studies, FSV UK, No. 260726.

**Institutional Review Board Statement:** Not applicable.

**Informed Consent Statement:** Not applicable.

**Data Availability Statement:** No new data were created or analyzed in this study. Data sharing is not applicable to this article.

**Conflicts of Interest:** The author declares no conflict of interest.

## Notes

1    *Mission des Augustins de l'Assomption*. January 1899. p. 181.
2    *Mission des Augustins de l'Assomption*. May 1902. p. 74.
3    *Mission des Augustins de l'Assomption*. May 1902. p. 75.
4    For instance, in 1902, four Assumptionists obtained the Greek rite with the permission of the Holy See. See *Mission des Augustins de l'Assommption*. May 1902. p. 75.
5    *Missions des Augustins de L'Assomption*. February 1897. p. 25.
6    L'Oeuvre de Vocation in *Echos du Noviciat de Notre Dame de l'Assomption* (Noviciat de Notre Dame de Livry). 1 January 1889. Paris: Imp.-gerant: E. Petithenry.
7    See Note 6.
8    The same school had students of various nationalities, 196 students in 1899 and 136 students in 1900; see Table 4 (Thobie 2009, p. 603).
9    *Mission des Augustins de l'Assomption*. May 1901. p. 326.
10   *Mission des Augustins de l'Assomption*. September 1901. p. 394.
11   For the exact numbers of girls enrolled in the three Oblate schools in Istanbul between 1893 and 1914, see (Thobie 2009, p. 674).

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
