# Peer review of "Nurturing Faith and Enlightening Minds: Assumptionist Education in the Ottoman Empire"

_religions, doi:10.3390/rel15010132_

Round 1

Reviewer 1 Report

Comments and Suggestions for Authors

The paper is well written. But it would have been better if the writer had brought in the experiences and responses of the local people of the empire to bear on the analysis. Leaving the local people's responses denies them agency in this interaction and the outcome.

There are some repetitions that should be removed.

Author Response

Dear Reviewer,

I appreciate your constructive feedback and guidance on my manuscript. Your insights have played a crucial role in enhancing the overall quality of the article. The comments provided have not only contributed to refining the current work but have also prompted a deeper reconsideration of my Ph.D. dissertation's focus on religious diplomacy between France, the Ottoman Empire, and the Vatican.

I have diligently addressed each of your suggestions and concerns, resulting in a comprehensive revision of the entire manuscript. Throughout this process, I have restructured and redistributed the content across various sections, incorporating new text to strengthen the overall narrative.

Efforts have been directed toward succinctly describing and contextualizing the theoretical background and empirical research, as well as improving the coherence, balance, and persuasiveness of the argument. The newly inserted text has been highlighted for enhanced clarity and visibility.

I believe these modifications significantly contribute to the manuscript's strength and cohesiveness. Your encouragement and guidance have been invaluable, and I am confident that the revisions made align with your expectations, further enriching the paper.

Thank you for dedicating your time and effort to reviewing my work. I eagerly anticipate your evaluation of the revised manuscript.

Best regards,

Reviewer 2 Report

Comments and Suggestions for Authors

I want to congratulate the author for this interesting study of the Assumptionist community. It is important that the study is written, in English in the context where most studies about the Assumptionist community are in French. So it provides a good opportunity to spread scientific information in English about the Assumptionist community created by the French Catholic missionary Emmanuel d'Alzon.

Positive aspects:

The article provides a good well-informed and well-argued historical narrative of the development of the Assumptionist communities and their relationship with the Ottoman Empire and Turkey. The article is clear, cursive in explanations, dynamic and has a rich bibliography. In addition, it can be observed that an important bibliography specific to the end of the 19th century and the beginning of the 20th century was studied, in order to more clearly define the historical context.

Recommendation:

1) I appreciate that the idea should be developed that during the studied period of the end of the 19th century and the beginning of the 20th century, the French language was equally the language of political diplomacy in Europe. This aspect represented an additional argument in the activity of the Assumptionist communities in Eastern Europe.

2) The second aspect is related to the author's intention to define, through the case study of the Assumptionist community, the cultural religious imperialism of France. I think it is an exaggeration to consider the missionary activity of the Assonist community as a way of promoting the cultural and religious imperialism of France ( r. 119-120: "Therefore, my objective is to utilize the Assumptionists as a case study to compre-119 hend the role of French Roman Catholic missionary education as a vehicle for France's 120 religiocultural imperialism."). I don't think the term imperialism is appropriate in this context and its use can be abandoned. It is well known that the quality work of the Assumptionist communities was from the beginning and is still now, to create bridges of communication and cooperation between the Christian churches of the East and the West. We also find this aspect in the conclusions of the study on lines 430-432, but it should be further developed. I believe that this lofty and generous objective for the educational quality of the Assumptionist communities should not be mixed with a political, economic and military objective (imperialism; r. 119-120; 126-127; 142, etc.).

Between lines 121-122 it is specified: "I intend to investigate the impact of Assumptionist education on various religious groups, including Catholics, Armenians, Greeks, and Muslims".

It is necessary to emphasize the fact that Armenians and Greeks are ethnic and national groups, not religious groups "per se", the aspects of religious affiliation must be explained more clearly

Author Response

Dear Reviewer,

I appreciate your feedback and guidance on my manuscript. Your insights have played a crucial role in enhancing the overall quality of the article. The comments provided have contributed to refining the current work and prompted a reconsideration of my Ph.D. dissertation's focus on religious diplomacy between France, the Ottoman Empire, and the Vatican.

I have diligently addressed each of your suggestions and concerns, resulting in a comprehensive revision of the entire manuscript. Throughout this process, I have restructured and redistributed the content across various sections, incorporating new text to strengthen the overall narrative. Efforts have been directed toward making the arguments and discussion of findings coherent, balanced, and compelling. The newly inserted text has been highlighted for enhanced clarity and visibility.

I believe these modifications contribute to the manuscript's strength and cohesiveness. Your guidance has been invaluable, and I am confident that the revisions made align with expectations, further enriching the paper.

Thank you for dedicating your time and effort to reviewing my work. I eagerly anticipate your evaluation of the revised manuscript.

Best regards,
